# Interval breast cancer is associated with other types of tumors

Felix Grassmann[1]*, Wei He[1], Mikael Eriksson[1], Marike Gabrielson[1], Per Hall[1,2] & Kamila Czene[1]

Breast cancer (BC) patients diagnosed between two screenings (interval cancers) are more likely than screen-detected patients to carry rare deleterious mutations in cancer genes potentially leading to increased risk for other non-breast cancer (non-BC) tumors. In this study, we include 14,846 women diagnosed with BC of which 1,772 are interval and 13,074 screen-detected. Compared to women with screen-detected cancers, interval breast cancer patients are more likely to have a non-BC tumor before (Odds ratio (OR): 1.43 [1.19–1.70], $P = 9.4 \times 10^{-5}$) and after (OR: 1.28 [1.14–1.44], $P = 4.70 \times 10^{-5}$) breast cancer diagnosis, are more likely to report a family history of non-BC tumors and have a lower genetic risk score based on common variants for non-BC tumors. In conclusion, interval breast cancer is associated with other tumors and common cancer variants are unlikely to be responsible for this association. These findings could have implications for future screening and prevention programs.

[1] Department of Medical Epidemiology and Biostatistics, Karolinska Institutet, Stockholm, Sweden. [2] Department of Oncology, Södersjukhuset, Stockholm, Sweden. *email: felix.grassmann@ki.se

Breast cancer screenings reduces breast cancer mortality by up to 35%[1,2]. However, in women regularly attending screening, between 20 and 30% of breast cancers are not detected by screening mammography but are diagnosed between screening intervals[3]. Failure of mammographic-screening protocols to detect cancer can be attributed to many different factors ranging from technical (e.g., imaging technology and radiologist interpretation)[4] to patient specific (e.g., age, co-morbidities, and breast/mammographic density)[5] to biological (e.g., the aggressive tumor grows to pathological, and palpable size within the screening window).

Compared with breast cancers detected at a screening, interval cancers are characterized by more-aggressive tumor characteristics and poorer prognosis (i.e., increased mortality) in most recent studies (summarized in ref. [6]). Patients with interval breast cancer usually present with an average higher histological grade[7,8], larger tumor size[9], and more-metastatic local lymph-nodes[10], a higher proportion of estrogen receptor (ER)[8]/progesterone receptor (PR) negativity[11], a higher frequency of HER2 positivity[8,9], and are more often triple negative[12].

Currently, few risk factors for interval breast cancers are known. Apart from high mammographic density[13] and current hormone replacement therapy (HRT)[12,14] use, a previous false-positive mammographic screening[15] as well as family history of breast cancer within first degree family members[14–16] have been implicated in interval cancer risk. Recently, we have shown that interval cancer cases (compared with screen-detected cancers) have a lower breast cancer genetic risk score (GRS), i.e., carry fewer breast cancer risk increasing alleles[17]. Conversely, rare loss of function mutations in 31 known and suspected breast cancer predisposition genes were found to be more common in interval cancer cases than in screen-detected cases[18].

Loss of function mutations in those genes are also frequently found to predispose individuals for other types of cancer[19] and may increase the incidence of other tumors in those individuals or in close relatives. In this study, we find that interval cancer patients are more likely than screen-detected patients to be diagnosed with other tumors, either before or after breast cancer diagnosis and that rather rare and not common variants are responsible for the observed the association. Our results thus reveal insights into risk and consequence of interval breast cancer.

## Results

**Investigating socio-economic and reproductive risk factors for interval cancer.** The current study included 14,846 breast cancer patients (1,772 interval cancer patients and 13,074 screen-detected cancer patients) from three cohorts (Table 1). Among known breast cancer risk factors, we found that interval compared with screen-detected breast cancer patients are more likely to have given birth before 25 years of age (OR: 1.13, 95% CI: 1.00–1.27, $P_{LR} < 0.05$, logistic regression, Wald test), are less likely to have a college degree (OR: 0.85, 95% CI: 0.77–0.96, $P_{LR} < 0.01$) and are more likely to report the use of hormone replacement therapy (OR: 1.29, 95% CI: 1.15–1.45, $P_{LR} < 0.001$) as well as a family history of breast cancer in close relatives (OR: 1.14, 95% CI: 1.01–1.29, $P_{LR} < 0.05$, Table 1). Importantly, interval cancers were consistently more likely than screen-detected cancers to have worse tumor characteristics and prognosis across the three cohorts (Table 1 and Supplementary Fig. 1).

**Prior and subsequent non-BC tumor diagnoses are associated with interval cancer.** We found that a prior non-BC tumor diagnosis was highly significantly associated with increased risk for interval compared with screen-detected cancer (OR: 1.43, 95% CI: 1.19–1.70, $P_{LR} = 9.4 \times 10^{-5}$, Fig. 1 and Table 1). Similarly, interval cancer cases compared to screen-detected cases were at higher risk to be affected with another non-BC tumor after breast cancer (OR: 1.28, 95% CI: 1.14–1.44, $P_{LR} = 4.70 \times 10^{-5}$, Fig. 1 and Table 1). Adjustment for age at first birth, education, BMI, HRT as well as family history of breast cancer did virtually not change the observed effect sizes and neither did the adjustment for percent mammographic density (which was unavailable in WHI) beyond the reduced sample size.

Next, we investigated which specific type of cancer diagnosis is associated with interval breast cancer compared with screen-detected breast cancer. We only investigated tumors which were observed in at least 30 patients across the whole cohort and found that a diagnosis of most cancer types is more common in interval compared with screen-detected breast cancer (Fig. 1). We found that a prior diagnosis of lung (OR: 2.57, 95% CI: 1.09–6.06, $P_{LR} <$ 0.05), colorectal (OR: 1.74, 95% CI: 1.02–2.97, $P_{LR} < 0.05$) or skin (OR: 1.70, 95% CI: 1.04–2.78, $P_{LR} < 0.05$) cancer was significantly associated with increased risk for interval breast cancer (Fig. 1). The increased risk due to skin cancer can be attributed to both, melanoma and non-melanoma skin cancers, although the association was not statistically significant for the latter. Likewise, compared with screen-detected breast cancer survivors, interval cancer survivors are significantly more likely to be diagnosed with colorectal cancer (OR: 1.48, 95% CI: 1.00–2.18, $P_{LR} < 0.05$) as well as skin (OR: 1.48, 95% CI: 1.04–2.11, $P_{LR} < 0.05$) cancer. However, here, the observed association of skin cancer with interval cancer risk was mostly restricted to non-melanoma skin cancers.

**Genetic insights into the association of interval breast cancer with non-BC tumors.** In order to elucidate potential genetic causes for the observed increased non-BC tumor prevalence in interval breast cancer patients, we investigated whether interval cancer patients are more likely to report a family history of other cancers compared with screen-detected patients. When we restricted the analyses to those individuals with any non-BC tumor diagnosis, we found that interval breast cancer patients compared with screen-detected patients are almost twice as likely to report a family history of non-BC tumors (OR: 1.94, 95% CI: 1.00–3.68, $P_{LR} < 0.05$, Table 2). This association was more pronounced in patients with a prior non-BC tumor diagnosis (OR: 3.41, 95% CI: 1.28–9.14, $P_{LR} < 0.01$) than in patients that were diagnosed with a tumor diagnosis after breast cancer (OR: 1.48, 95% CI: 0.89–3.52, $P_{LR} > 0.05$).

The increased prevalence of non-BC tumor diagnoses in interval compared with screen-detected breast cancers can potentially be attributed to common genetic risk factors. In line with prior studies[20], we found that interval cancer patients compared with screen-detected patients have a significantly lower breast cancer genetic risk score in the combined study (OR per standard deviation (SD): 0.85, 95% CI: 0.76–0.95, $P_{LR} < 0.01$, Fig. 2). Next, we computed 12 genetic risk scores for different cancers (Fig. 1) from a total of 304 genome-wide significant cancer risk increasing variants and an all non-BC tumor genetic risk score by calculating the sum of the individual cancer scores. We excluded all variations in published, genome-wide significant breast cancer loci in order to exclude potential pleiotropic effects of known BC variants on other cancers (Supplementary Data 1 and Supplementary Fig. 2). As expected, all of the cancer-specific genetic risk scores were not positively correlated to the breast cancer score, indicating that they likely capture common cancer genetics independent of known BC genetic variance (Supplementary Fig. 2). Similar to the breast cancer genetic score, we found that interval breast cancer cases had a significantly lower number of non-BC tumor risk increasing alleles (OR per SD of

**Table 1 Summary statistics and association analysis of risk factors and tumor characteristics**

| Variable | KARMA IC | KARMA SDC | KARMA IC vs. SDC[a] | LIBRO-1 IC | LIBRO-1 SDC | LIBRO-1 IC vs. SDC[a] | WHI IC | WHI SDC | WHI IC vs. SDC[a] | Joint analysis[b] |
|---|---|---|---|---|---|---|---|---|---|---|
| Number of individuals | 395 | 1857 | 2252 | 224 | 1698 | 1922 | 1153 | 9519 | 10,672 | NA |
| Age at diagnosis (SD) (years) | 59.15 (8.00) | 59.32 (8.29) | 0.997 (0.984, 1.011) | 60.43 (5.61) | 59.63 (5.78) | 1.025 (1.000, 1.051)* | 70.93 (8.18) | 71.22 (7.73) | 0.997 (0.989; 1.005) | 0.999 (0.992; 1.005) |
| Age at menarche (SD) (years) | 13.10 (1.39) | 13.19 (1.46) | 0.959 (0.886, 1.036) | 13.25 (1.45) | 13.18 (1.47) | 1.021 (0.924, 1.127) | 12.52 (1.45) | 12.53 (1.46) | 1.006 (0.964; 1.050) | 0.992 (0.958; 1.027) |
| Age at first birth < 25 years (%) | 43.07 | 40.96 | 1.093 (0.860, 1.388) | 42.59 | 47.97 | 1.229 (0.843, 1.791) | 61.94 | 58.91 | 1.120 (0.971; 1.294) | 1.128 (1.005; 1.268)* |
| Education level (% college or university degree) | 43.09 | 47.83 | 0.825 (0.655, 1.037) | 52.17 | 56.54 | 0.825 (0.575, 1.184) | 70.74 | 73.77 | 0.877 (0.765; 1.007) | 0.854 (0.765; 0.955)** |
| Body mass index closest to diagnosis (SD) (kg/m²) | 26.15 (3.94) | 26.13 (4.38) | 1.001 (0.975, 1.027) | 25.58 (4.39) | 25.79 (4.13) | 0.987 (0.952, 1.021) | 28.47 (5.93) | 27.99 (5.68) | 1.012 (1.001; 1.023)* | 1.009 (0.999; 1.018) |
| Hormone replacement therapy at diagnosis (%) | 37.93 | 28.30 | 1.376 (1.157, 1.630)*** | 59.16 | 48.25 | 1.287 (1.066, 1.550)*** | 57.59 | 50.7 | 1.310 (0.905; 1.906)c | 1.288 (1.147; 1.446)*** |
| Current smoker at interview (%) | 10.13 | 13.27 | 0.952 (0.805, 1.122) | 8.72 | 11.73 | 0.925 (0.721, 1.179) | 7.66 | 5.88 | 1.048 (0.946; 1.159) | 1.026 (0.848; 1.234) |
| Non-BC tumor prior to BC diagnosis (%) | 6.83 | 4.47 | 1.469 (0.985, 2.134)* | 5.80 | 3.95 | 1.510 (0.821, 2.583) | 4.42 | 2.99 | 1.421 (1.136; 1.752)** | 1.425 (1.187; 1.695)*** |
| Non-BC tumor after BC diagnosis (%) | 9.37 | 5.22 | 1.692 (1.232, 2.305)*** | 15.62 | 11.66 | 1.440 (1.103, 1.866)** | 10.23 | 8.77 | 1.145 (0.983; 1.324) | 1.273 (1.131; 1.427)*** |
| Family history of BC (%) | 22.69 | 19.52 | 1.211 (0.916, 1.589) | 22.73 | 17.12 | 1.399 (0.986, 1.955) | 25.83 | 23.47 | 1.197 (0.807; 1.748) | 1.143 (1.014; 1.286)* |
| Tumor size (SD) (cm) | 2.10 (1.36) | 1.58 (1.10) | 1.372 (1.232, 1.530)*** | 1.87 (1.21) | 1.60 (1.16) | 1.184 (1.061, 1.315)** | 1.75 (1.7) | 1.53 (1.27) | 1.08 (1.062; 1.155)*** | 1.013 (1.010; 1.017)*** |
| Number of positive lymphnodes (SD) | 1.42 (2.98) | 0.64 (1.93) | 1.132 (1.075, 1.193)*** | 1.03 (2.10) | 0.63 (1.74) | 1.098 (1.026, 1.171)*** | 6.62 (7.92) | 5.76 (7.06) | 1.015 (1.006; 1.023)*** | 1.019 (1.011; 1.027) |
| Distant metastasis (%) | 0.94 | 0.08 | 12.043 (1.145, 260.31)* | 0.93 | 0.00 | N/A | 2.6 | 0.8 | 3.329 (2.139; 5.050)*** | 3.666 (2.408; 5.468)*** |
| Positive ER status (%) | 85.20 | 90.14 | 0.626 (0.418, 0.961)* | 76.35 | 91.16 | 0.313 (0.206, 0.485)*** | 82.34 | 85.19 | 0.834 (0.700; 0.998)* | 0.722 (0.621; 0.841)*** |
| Positive PR status (%) | 62.96 | 76.09 | 0.536 (0.395, 0.731)*** | 54.93 | 74.54 | 0.417 (0.292, 0.596)*** | 67.75 | 73.07 | 0.783 (0.678; 0.907)*** | 0.686 (0.607; 0.775)*** |
| Positive HER2 status (%) | 19.10 | 10.56 | 2.071 (1.337, 3.147)*** | 20.00 | 10.11 | 2.134 (0.808, 5.024) | 19.35 | 13.47 | 1.518 (1.240; 1.848)*** | 1.624 (1.357; 1.934)*** |
| Triple negative (%) | 8.23 | 6.04 | 1.397 (0.734, 2.494) | 20.59 | 5.68 | 4.724 (1.708, 11.994)*** | 11.56 | 9.3 | 1.233 (0.958; 1.568) | 1.320 (1.053; 1.640)*** |
| Histological grade >I (%) | 89.09 | 76.30 | 2.585 (1.692, 4.121)*** | 88.54 | 75.03 | 2.566 (1.405, 5.163)*** | 76.61 | 74.51 | 1.109 (0.953; 1.294) | 1.296 (1.128; 1.493)*** |

aStatistical evaluation of differences between IC and SDC in individual studies using logistic regression, additionally adjusted for study.
bStatistical evaluation of differences between IC and SDC in the whole-study using logistic regression. The odds ratio for IC and the respective 95% confidence intervals (square brackets) are reported per unit for continuous variables. Adjusted for age at diagnosis (and self-reported race in the WHI).
cOnly evaluated in patients within the HRT clinical trial arm with BC diagnosed during the trial period. Adjusted for average adherence rate before BC diagnosis
IC interval breast cancer, SDC screen-detected breast cancer; *P < 0.05; **P < 0.01; ***P < 0.001; SD standard deviation, ER estrogen receptor, PR progesterone receptor

the all non-BC tumor genetic risk score: 0.88, 95% CI: 0.80–0.96, $P_{LR} < 0.01$, Fig. 2) compared with screen-detected patients. We also investigated if this effect can be attributed to any cancer-specific genetic score (Fig. 2) and found that the genetic risk score for skin cancer was significantly associated with reduced risk for interval breast cancer compared with screen-detected breast cancer (OR per SD: 0.91, 95% CI: 0.83–0.99, $P_{LR} < 0.05$). Similar to the results observed from the actual tumor diagnoses, we found that the risk score for non-melanoma skin cancer was more strongly associated with reduced disease risk than the melanoma-specific risk score. Importantly, most of the other genetic risk scores are also protective for interval compared with screen-detected cancer.

Finally, we investigated the association of individual cancer risk variants with interval breast cancer risk (Supplementary Data 1). Although several variants were nominally significantly associated with interval breast cancer ($P_{LR} < 0.05$), after adjustment for multiple testing (Bonferroni correction), none of the variants remains statistically significantly associated.

## Discussion

In this study, we showed that non-BC tumors, either before or after BC diagnosis, are associated with interval breast cancers and that in breast cancer patients with a second primary tumor diagnosis, interval compared with screen-detected patients are three times more likely to report a family history of other cancers. In addition, we demonstrated that interval cancer patients compared with screen-detected patients have fewer common cancer risk increasing alleles of both, breast cancer, and other cancers.

Our results are in line with recent findings that interval breast cancer patients compared with screen-detected patients have fewer common breast cancer risk increasing alleles and, on the other hand, are more likely to carry rare deleterious (loss of function) mutations in known or suspected cancer genes[17,18]. Although previous efforts to identify cancer genes have largely focused on single tumor types, recent evidence suggests a pleiotropic role those genes[21] owing to their involvement in fundamental processes required for tumor initiation and propagation such as genomic integrity[22,23], hormonal processes, angiogenesis, and inflammation[24]. As such, mutations in those genes frequently predispose for different cancers and cancer syndromes[21,25] and might result in multiple tumor diagnoses over an individual's lifetime as well as increase familial risk for tumors. Importantly, patients with germline mutations in cancer genes are often diagnosed with more-aggressive and fast growing tumors[26–28] and thus may more likely be diagnosed with interval than screen-detected breast cancer. Nevertheless, mutations in currently known cancer genes can only explain a small portion of the observed association. Thus, further large-scale sequencing efforts in patients with multiple tumors are necessary to uncover the genetic basis for both, multiple cancer diagnoses and increased interval breast cancer risk.

In addition to the overall increased occurrence of other cancers in interval compared with screen-detected breast cancer patients, we found that virtually all types of cancer are more common in interval compared with screen-detected cancer patients. Although we only found a statistically significant increase of prior lung, skin, and colorectal as well as subsequent skin and colorectal tumors in interval compared with screen-detected cases, the lack of significance for the other cancers can be mainly attributed to the limited number of interval breast cancer patients with other tumor diagnoses in our data set. Interestingly, we did not find an increased number of ovarian cancer survivors in interval compared with screen-detected breast cancer patients, although both, breast and ovarian cancer, have well documented shared genetic

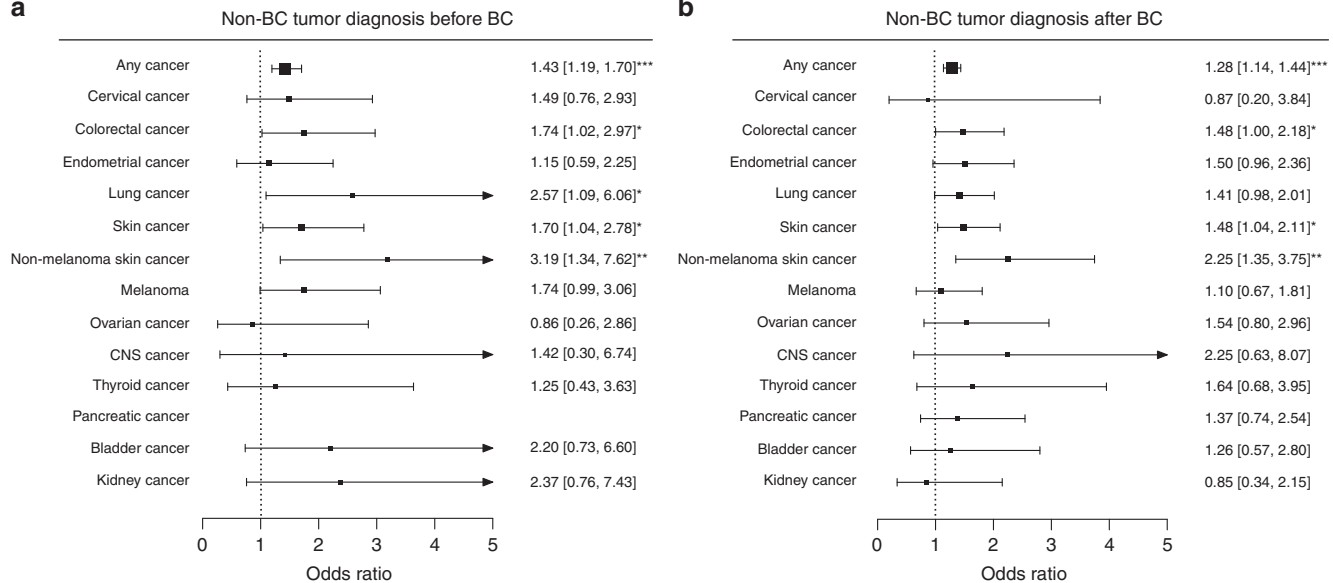

**Fig. 1** Prior and subsequent non-BC tumor diagnoses in IC compared with SDC. The effect size estimates of the association with interval breast cancer (IC) risk (black squares) compared with screen-detected breast cancer (SDC) as well as the 95% confidence intervals (CI, horizontal lines) for different prior and subsequent cancers are shown. The exact estimates derived from logistic regression models adjusted for age at diagnosis are given on the right-hand side of the plot with the accompanying 95% CI. **a** A prior non-BC tumor diagnosis (any type) as well as a prior lung, skin, non-melanoma skin, or colorectal cancer diagnosis was significantly associated with increased risk for IC compared with SDC. **b** A non-BC tumor diagnosis after breast cancer (any type) as well as a subsequent colorectal, skin, or non-melanoma skin cancer diagnosis was significantly more common in IC cases than in SDC cases. *P < 0.05; **P < 0.01; ***P < 0.001

---

**Table 2 Association of non-BC family history with interval breast cancer**

| Other cancer status[a] | Family history in IC patients (%) (N)[b] | Family history in SDC patients (%) (N)[b] | OR (95% CI)[c] |
|---|---|---|---|
| All patients | 14.10 (454) | 15.53 (2280) | 0.935 (0.691, 1.249) |
| Any non-BC tumor | 24.05 (79) | 16.26 (289) | 1.935 (1.001, 3.675)* |
| Any prior non-BC tumor | 37.93 (29) | 16.04 (106) | 3.411 (1.284, 9.140)** |
| Any non-BC tumor after BC | 17.31 (52) | 15.71 (191) | 1.484 (0.589, 3.523) |

[a] Stratification of patients according to non-BC tumor status
[b] Family history of tumors other than breast cancer was only available in WHI and KARMA
[c] Odds ratio and 95% confidence intervals (in brackets) of the assocation of non-BC tumor family history with interval breast cancer compared with screen-detected cancer in Caucasians. The logistic regression models were adjusted for age at diagnosis, study and the all non-BC cancer GRS
IC interval breast cancer, SDC screen-detected breast cancer, *P < 0.05; **P < 0.01, N number of individuals with non-missing family history information

---

risk factors[29,30]. However, those genetic factors only explain a fraction of the breast cancer cases in ovarian cancer survivors[31]. Therefore, in order to find such an association, we would have to restrict our analysis to younger cases with multiple cancer diagnoses in their family, which is not possible owing to limited number of individuals that survived ovarian cancer in our data set. Alternatively, removal of both the uterus and the ovaries/fallopian tubes as part of ovarian cancer treatment may change sex hormone levels favorably for interval cancer risk, which is likely linked to hormonal changes owing to its risk association with hormone replacement therapy. The increased number of non-BC tumors in interval cancer cases compared with screen-detected cases can potentially be influenced by several factors. For instance, more-aggressive treatment of interval cancer patients[32,33] could result in increased tumor incidence. However, as noted above, we found an increased occurrence of almost all particular cancers, thus, disqualifying treatment effects from being a major influence on the association[34,35].

Rare mutations that are associated with a strong increase in risk and a worse prognosis in breast cancer are more frequent in interval cancer patients than in screen-detected patients[18] and are also often aggregated in families with a (breast) cancer

history. Accordingly, we found that family history of non-BC tumors is associated with increased risk for interval compared with screen-detected cancer (and consequently with worse prognosis) in patients with another tumor diagnosis. Importantly, the association of family history of non-BC tumors with increased interval cancer risk was found in patients with either a prior or a subsequent tumor. The association of non-BC family history with interval cancer was markedly stronger in patients with a prior compared with a subsequent non-BC tumor diagnosis. The reduced effect size can potentially be attributed to the increased incidence of secondary non-BC tumors that arise by chance with advanced age and are not linked to a strong rare genetic predisposition but rather shared environmental or behavioral factors, diluting the observed association[36,37].

The finding that both, common breast cancer and non-BC risk increasing alleles, are less frequent in interval compared with screen-detected cancer provides insights into the genetic basis of interval breast cancer. A possible explanation for common breast cancer alleles being associated with screen-detected (less aggressive) tumors may be attributed to the study population in which the common variants were identified. Most breast cancer patient

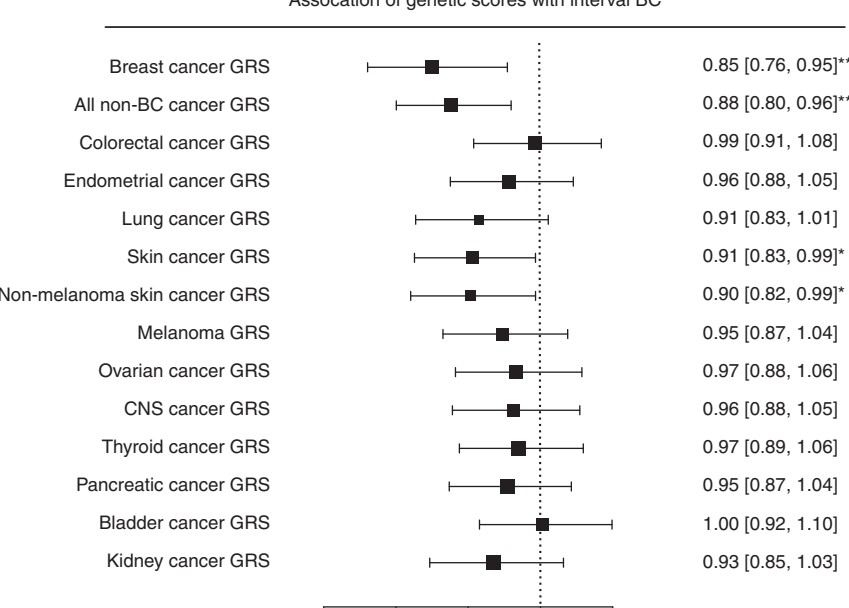

**Fig. 2** Cancer genetic risk scores in interval breast cancer risk. Cancer genetic risk scores (GRS) were computed in Caucasian patients from the LIBRO-1 (N = 1784), KARMA (N = 1690) and WHI (N = 1585) studies. The odds ratios of different cancer genetic risk scores (GRS) on interval breast cancer (IC) risk (black squares) compared with SDC as well as the 95% confidence intervals (CI, horizontal lines) were computed with logistic regression, adjusted for age at diagnosis, study and the first three principal components. The exact estimates are given on the right-hand side of the plot with the accompanying 95% CI. The breast cancer GRS, skin cancer GRS, non-melanoma skin cancer GRS as well as the combined all cancer GRS computed from the cancer risk increasing alleles of 304 variants (excluding any variants in known BC loci) was statistically significantly associated with a protective effect on IC risk compared with SDC. The other cancer-specific GRS were not significantly associated, although the majority of scores had a protective effect on IC. The GRS effect sizes are given per standard deviation of the score. *$P < 0.05$; **$P < 0.01$

cohorts are likely enriched for screen-detected cancers and breast cancer survivors. In addition, most of the patients in the cohorts which are part of the Breast Cancer Association Consortium (BCAC)[38] have ER-positive tumors and thus, a large proportion of the previously reported common variants may show a stronger association with ER-positive breast cancers, which are in turn associated with more benign tumors compared with ER-negative breast cancers. In addition, those genome-wide association studies were not designed to identify interval cancer-specific variants and thus the aggregate score would not be expected to reflect such an association. However, here, we show that other cancer risk increasing alleles are also less frequent in interval compared with screen-detected cancer patients, implicating that common cancer risk increasing alleles, in general, result less aggressive (screen-detected) breast cancer. Similar to the notion that the BCAC cohorts are more likely to include less-aggressive breast cancer cases, large case–control cohorts for other cancers may also be enriched for cancer survivors with less severe phenotypes. Therefore, genetic risk scores based on those association results may also predict milder phenotypes and more favorable prognosis. Alternatively, although we excluded variants in known BC loci in order to reduce the influence of known BC risk variation in the current analysis, it is possible that there is pleiotropy with BC present in the other cancer genetic risk scores, which is not captured by known BC risk variants and could potentially drive the association.

In order to expand the number of available studies with interval breast cancer status, this, to our knowledge, is the first study to compute incident interval breast cancer status from questionnaire in the Women's Health Study. Although the precise screening history and mammographic density estimates were not

available in the WHI, compared with both Swedish studies, we observed that the interval cancer cases derived from questionnaire data have a comparable risk factor and tumor characteristic distribution as well as a similar prognosis. The effect sizes were, however, generally weaker in the Women's Health Initiative (WHI) than in the Swedish cohorts and several factors may explain this observation. First, misclassification of interval cancers in the WHI is possible because we computed incident interval breast cancer status from questionnaire rather than using precise screening history. Second, the WHI did not report density measurements, thus preventing us from excluding intervals cancers missed at screening owing to masking effects of high mammographic density. Third, the increased age of the WHI participants may have resulted in increased mortality and thus lower incidence of other tumors after breast cancer diagnosis. However, these limitations are likely to dilute rather than inflate the observed results, as restricting our main analysis to individuals from LIBRO-1 and KARolinska MAmmography project for risk prediction of breast cancer (KARMA) (i.e., studies which have actual prior screening dates) yielded consistent results (Supplementary Fig. 3).

The current studies aimed at providing insights into interval cancer (genetics) in order to further characterize tumors developing in women not currently benefiting from regular mammographic screenings. Thus, the current results are not intended to be directly relevant for clinical practice beyond raising awareness that some of these patients and family members are at increased risk for interval breast cancer and other tumors. Although the increased risk for interval cancer in cancer survivors may have potential implications for screening or prevention programs, other risk factors, co-morbidities as well as their potential impact

on prognosis need to be considered jointly in disease management.

One of the limitations of our current study is that the types of tumor reported in close family members (i.e., family history of other cancers) in KARMA and WHI differed. In KARMA, participants were asked about ovarian and pancreatic cancer in close relatives while in WHI any type of cancer in close family members was ascertained. Furthermore, the variable was missing from the LIBRO-1 study. In addition, the frequency of estrogen receptor-positive tumors is somewhat higher in our IC cases compared with the frequencies observed in IC patients from other studies[39–41], which can be attributed to different periods of inclusion and differences in recruitment strategy. However, those differences should not affect the main conclusions of this study. Finally, in this study we have not identified the actual cause of the increased frequency of other tumors in interval compared with screen-detected patients and thus were only able to speculate on potential mechanisms. Among the strengths of this study are its large sample size with a broad range of available risk factor, tumor characteristics, and mortality data, precise ascertainment of other cancers before and after BC diagnosis, the inclusion of breast cancer patients from different study designs (population based and prospective study) as well as the inclusion of both, pre- and post-menopausal women.

In conclusion, we show that either a previous or a subsequent non-BC tumor diagnosis was more common in interval compared with screen-detected breast cancer in three independent cohorts and that common cancer risk increasing variants are unlikely to be responsible for this association. The association of family history of other cancers with interval cancer risk suggests that rare penetrant cancer mutations predispose individuals for both, interval breast cancer and other cancers and further large-scale sequencing efforts (especially in patients with more than one tumor) are necessary to uncover the underlying cause of the observed associations. Our results thus reveal insights into risk and consequence of interval breast cancer and highlight further genetic differences between interval and screen-detected breast cancer, which could have significant implications for future screening programs and the clinical management of cancer survivors.

## Methods

**Study population.** Ethical approvals of the KARMA and LIBRO-1 studies were given by the ethical review board at Karolinska Institutet (Stockholm, Sweden) and written informed consent was obtained from all participants. Study approval for data analysis in the Women's Health Initiative (WHI, ClinicalTrials.gov Identifier: NCT00000611) was granted by the Regional Ethical Review Board in Stockholm, Sweden. Data from the WHI were retrieved from the Database of Genotypes and Phenotypes (dbGAP, URL: https://www.ncbi.nlm.nih.gov/gap) under the accession number *phs000200.v11.p3*, with approval from the data-access committee.

KARMA is a mammography screening-based study initiated in January 2011 and includes women who attended mammography screening or clinical mammography at four hospitals in Sweden[42,43]. As the date of breast cancer diagnosis, we only counted the first occurrence of breast cancer. Breast cancer risk factors (such as family history of breast cancer and other cancers, body mass index, smoking status, and hormone replacement therapy at diagnosis), socio-economic factors as well as reproductive events were ascertained from mailed questionnaire. In addition, tumor characteristics were retrieved from the Swedish cancer registry and survival after breast cancer was ascertained from the Swedish causes of death registry, as described before[43] with virtually no missing data.

LIBRO-1 is a breast cancer cohort with 5715 patients who gave informed consent and were diagnosed between 2001 and 2008 in the Stockholm/Gotland area. Similar to the KARMA study, breast cancer risk factors, socio-economic factors, and reproductive events were assessed by questionnaire and tumor characteristics as well as cause and date of death were retrieved from the respective Swedish registries. However, family history of tumors other than breast cancer was not available in LIBRO-1.

In order to expand the number of patients for our analysis and to investigate our findings in a large collection of incident breast cancer patients, we included breast cancer patients from the Women's Health Initiative. The WHI is a prospective national health study of >160,000 post-menopausal women aged 50–79

recruited between 1993 and 1998 at 40 clinical sites in the United States[44,45], with an average follow-up time of 17.20 (S.D. 4.17) years.

The study consists of two major parts: (1) the observational study without intervention, and (2) the clinical trial arm which investigated the role of hormone replacement therapy, dietary modification, and vitamin supplementation on different outcomes such as cancer, cardiovascular disease, and osteoporosis. We identified 11,031 women with incident BC, whose BC status was ascertained by medical review (and excluded BC cases that were ascertained by death certificate). All demographic and socio-economic variables, breast cancer risk factors, and tumor characteristics were extracted from the respective aggregate outcome or questionnaire tables as deposited in dbGAP. The breast cancer patients in the WHI were comprised of different ethnicities: 87.7% were self-reported Caucasians, 6.8% were Blacks or African-Americans, 2.0% were Asians and 2.2% were Hispanics/Latinos, whereas 0.9% reported to belong to another ethnic group. We did not observe significant a difference in interval cancer incidence between ethnicities and thus analyzed all BC cases in WHI jointly while adjusting the statistical analyses for self-reported ethnicity.

**Interval cancer ascertainment.** In both, the KARMA and LIBRO-1 study, interval cancer status was ascertained as recently described[12,46]. In brief, all women in the Stockholm area aged 50–69 have been invited to participate in regular mammographic screenings every 24 months since 1989, whereas women aged 40–49 were included from mid-2005 and screened at 18-month intervals. A BC diagnosed after a negative mammographic screening before the next scheduled screening visit was considered interval breast cancer. Conversely, cancers diagnosed at a regularly scheduled mammographic screening were deemed screen-detected cancer. Patients who did not attend the screening mammogram prior to breast cancer diagnosis (i.e., their last screening mammogram was >18/24 months before breast cancer diagnosis) or with missing prior screening information were excluded from our data sets. We also determined the mammographic density of the prior screening mammogram (i.e., the last mammogram preceding breast cancer diagnosis) with STRATUS, a machine-learning algorithm capable of computing density from both, analog and digital mammograms[47]. Patients without data on mammographic density (e.g., owing to missing mammographic images) were excluded from the KARMA and LIBRO-1 study. Furthermore, we also excluded interval cancer patients (but not screen-detected patients) with medium or high mammographic density (percent mammographic density > 25), as the tumor might have been masked by the high dense tissue and thus not been detected at the prior screening mammogram. Accordingly, this collection of interval breast cancer cases should represent 'true' interval cancer cases not confounded by missed screen-detected tumors.

In KARMA, out of 3862 BC patients, we excluded 287 patients who did not attend their scheduled screening mammogram prior to the BC diagnosis, 680 patients with no history of screening prior to BC diagnosis and 216 patients with missing mammographic density estimates. We also excluded 427 out of 822 (52%) interval cancer patients with medium or high mammographic density. In total, 2252 BC patients (either interval cancer cases or screen-detected cases) were included in the present study (Table 1). Similarly, in LIBRO-1, we excluded 1014 patients which did not attend their last screening, 1800 patients, which never attended screening before breast cancer or were diagnosed before their first invitation, 562 patients with missing mammographic density information as well as 417 out of 641 (65%) interval cancer cases with medium or high dense breasts (percent mammographic density > 25). In total, 1922 women with either interval or screen-detected BC were included in the present study (Table 1).

In the WHI, interval cancer status was determined from the self-reported mammography questionnaire data ascertained annually by mail or interview. In case a woman reported a mammographic screening since the last follow-up, we assumed that the mammogram was taken in the middle of this interval (i.e., between the present and the previous follow-up questionnaire) and compared those screening dates with the date of breast cancer diagnosis. In order to account for imprecision in determining the accurate screening date, we defined a breast cancer diagnosed within 3 months of an estimated screening date as screen-detected cancer, whereas breast cancers diagnosed between three and 24 months after the last screening were considered interval cancers. We also excluded patients without information on their mammographic screening history ($N = 370$) as well as those who did not report attending a mammographic screening more than 2 years prior to breast cancer diagnosis. ($N = 259$). In total, 10,672 post-menopausal breast cancer patients from both, the observational and clinical trial arms were included in the current study (Table 1).

Utilizing our approach, we found that there were 87.26 breast cancer diagnoses (8.9 interval and 73.49 screen-detected) per 10,000 self-reported mammographic screens, in line with numbers typically observed in screened populations[6]. The average time between reported screenings was 581.69 days (S.D. 350.10), in line with a mixture of women screened annually and biennially. The average time between prior mammogram and interval cancer diagnosis was 243.89 days (S.D. 163.32). Thus, 82% of the identified interval cancer cases were discovered within one year after the last screening in WHI compared with 42 and 33% in KARMA and LIBRO-1, respectively. However, previous research has shown that there is little difference in tumor characteristics between interval breast tumors detected within the first or second year of the screening regimes[6,12]. In general, the women

in the WHI were diligent in answering their yearly questionnaire with an average time between yearly questionnaires of 383.92 days (S.D. 77.03)

**Assessment of non-BC tumor diagnosis**. For both, KARMA and LIBRO-1 participants, extensive registry data are available. Thus, for breast cancer patients in those studies, we extracted the number of (prior and subsequent) non-BC tumor diagnoses from the Swedish cancer registry. We only counted (solid) cancer diagnoses within 15 years prior to breast cancer diagnosis and available ICD10 code. We did not include women diagnosed with prior non-BC tumor diagnoses less than 1.5 years (540 days) before breast cancer diagnosis, as the breast cancer might be diagnosed in the diagnostic workup of the previous cancer[48] and thus be incorrectly detected as interval cancer. In addition, we calculated the occurrence of specific (common) cancers in women: lung (ICD10 code: C34), cervical (C53), endometrial (C54), colorectal (C18 and C20), skin (C43 and C44), non-melanoma skin cancer (C44), melanoma (C43), ovarian (C56), thyroid (C73), central nervous system (C69–72), pancreas (C25), bladder (C67), or kidney cancer (C64).

All 10,672 post-menopausal BC patients in WHI were incident breast cancer cases and we counted the number of (centrally adjudicated/evaluated) other, solid non-BC tumor diagnoses between study enrollment and the first breast cancer diagnosis and those that occurred after the first BC diagnosis. In the WHI, only melanoma diagnoses were available and thus skin cancer only included melanoma cases and no non-melanoma skin cancers. Similarly, we documented the diagnosis of the same specific tumors as above and did not consider women with non-BC tumor diagnoses that occurred within 1.5 years before BC diagnosis.

**Genotyping and imputation**. Patients in KARMA and LIBRO-1 were genotyped through the BCAC on a custom Illumina iSelect genotyping array as part of the Collaborative Oncological Gene-environment Study (iCOGS) or on the OncoArray. The genotyping data used for this study from the KARMA project was first described in ref. [38] for 398 individuals genotyped on the iCOGS array and in ref. [49] for 1292 women genotyped on the OncoArray platform. Similarly, 1784 women in LIBRO-1 (as part of the pKARMA study) were genotyped on the iCOGS array which was first described in ref. [38]. Quality control and imputation of missing and un-genotyped variants was performed by the BCAC. In brief, missing genotypes were imputed to the 1000 Genomes Phase 3 reference haplotypes using ShapeIt[50] and IMPUTE[51] and variants with an imputation quality ($R^2$) > 0.8 and a minor allele frequency > 0.01 were retained.

Patients from the WHI were genotyped on different genotyping platforms as part of various sub-studies and were jointly imputed to the 1000 Genomes Phase 3 reference by the WHI investigators. Imputed genotype data were retrieved from dbGAP (accession: phs000746) for 971 out of 9103 self-identified Caucasian BC cases[52] in the minimac[53] dosage/matrix format and converted to variant call format (vcf) in R[54]. The genotypes were coded as the expected genotype dosage (DS) of each non-reference (ALT) allele. We excluded variants that were poorly imputed ($R^2 < 0.3$) or which showed a significant deviation from Hardy–Weinberg–Equilibrium (HWE, $P < 1.00 \times 10^{-6}$, HWE exact test). We also extracted the un-imputed genotypes of an additional 614 BC cases genotyped on the Metabochip as part of the PAGE WHI study[55] (accession: phs000227) and imputed missing genotypes using the 1000 Genomes Phase 3 (version 5) reference haplotypes. First, we excluded very rare and poorly performing genotyped variants (deviation from HWE ($P < 0.0001$, HWE exact test), minor allele frequency < 0.001, missing rate per variant > 30%) and poorly genotyped samples (missing rate overall >5%). Next, the phase of each variant was determined with ShapeIt2[51] and missing genotypes were imputed with minimac3[56] with standard settings. We excluded poorly imputed variants with the same criteria as above. Finally, we merged all imputed genotype sets with bcftools[57] to generate a single genotype file for all genotyped samples.

In order to account for potential population stratification and mixed ethnicities, we computed the first three principal components from the relatedness matrix derived from imputed genotypes with QCTOOL (http://www.well.ox.ac.uk/~gav/qctool) for all studies.

**Genetic risk scores calculation**. We calculated the genetic risk score as the effect size (log odds ratio) weighted sum of cancer risk increasing alleles, normalized to the average effect size of all variants[17,58,59]. Thus, an increase in one point of the genetic risk score corresponds to having one additional risk allele with average effect size. For the BC genetic risk score, we included 158 variants associated with breast cancer risk with genome-wide significance ($P < 5.00 \times 10^{-8}$), as recently reported by BCAC (Supplementary Data 1)[60].

In addition, we computed genetic risk scores for 12 different, common cancers from genome-wide significant variants (identified primarily in Caucasian/European populations) retrieved from the GWAS catalog[61] (version 1.0.2, accession date: 25 June, 2018, Supplementary Data 1). We excluded variants that are located in known, genome-wide significant breast cancer loci[60]. A known breast cancer locus was defined by the most distant variants in moderate linkage disequilibrium ($R^2 > 0.5$) with the genome-wide significant lead variant and an additional 500,000 base-pairs added up- and downstream. In particular, we computed a non-BC genetic risk score for colorectal cancer (35 variants), endometrial cancer (5 variants), lung cancer (78 variants), skin cancer (57 variants), melanoma (18 variants), non-melanoma skin cancer (39), ovarian cancer (29 variants), bladder cancer (11

variants), thyroid cancer (11 variants), central nervous systems (36 variants), renal cancer (15 variants), and pancreatic cancer (14 variants). We did not compute a genetic score for cervical cancer as there were less than five genome-wide significant variants in Europeans reported as of 25 June, 2018. For the all non-BC cancer genetic risk score, we computed the sum of the individual genetic risk scores, effectively generating a compound score of 304 variants associated with 12 different cancers, excluding variations in known breast cancer susceptibility loci (see Supplementary Data 1).

**Statistical analyses**. All statistical analyses were performed in R[54]. We used logistic regression, adjusted for age at diagnosis to identify factors that were significantly different between screen-detected and interval cancers. In addition, in the joint/pooled analyses, we additionally adjusted for study. The reported P values of association from the logistic regression models were based on a Wald test and are denoted as $P_{LR}$. All genetic analyses were restricted to individuals from European descent (Caucasians) and additionally adjusted for the first three principal components of ancestry computed from the imputed genotypes.

In order to estimate the effect of interval cancer diagnosis on overall (all cause) and/or breast cancer-specific survival, we fit Cox proportional hazard models (adjusted for age at diagnosis) as implemented in the survival package[62]. We plotted the results with the ggadjustedcurves function (from the package survminer[63] implemented in R).

**Reporting summary**. Further information on research design is available in the Nature Research Reporting Summary linked to this article.

## Data availability

Genotypes and phenotypes from the Women's Health Initiative is available through dbGAP (accession phs000200.v11.p3). Access to phenotypes, biospecimen and genotypes from the KARMA study can be requested from https://karmastudy.org/data-access/. Access to the LIBRO-1 phenotypes and genotypes is restricted due to IRB requirements but data can be shared upon reasonable request to the PIs of LIBRO-1 (Kamila Czene and Per Hall).

## Code availability

The analysis script to compute risk factors, tumor characteristics as well as interval cancer status in the WHI based on the phenotypic information and questionnaire data retrieved from dbGAP is available at https://github.com/GrassmannLab/IC_WHI.

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

## Acknowledgements

This work was financed by the Swedish Research Council (Grant 2018-02547), the Swedish Cancer Society (grants CAN 2016/684 and 2013/469), the Cancer Society in Stockholm (Grant 141092), the Stockholm County Council (Grant No. LS 1211–1594), and the Karolinska Institutet's Research Foundation (Grant 2018-02146). The KARMA study is supported by the Märit and Hans Rausing Initiative Against Breast Cancer and the Cancer and Risk Prediction Center (CRisP), a Linnaeus center (grant 70867902) financed by the Swedish Research Council. F.G. was a Leopoldina Postdoctoral Fellow (Grant No. LPDS 2018-06) funded by the Academy of Sciences Leopoldina. W.H. was supported by the Swedish Research Council for Health, Working Life and Welfare (FORTE, 2018-00877). We thank Dr. Johanna Holm for her valuable contribution to the manuscript. Genotyping of the OncoArray was principally funded by three sources: the PERSPECTIVE project, funded by the Government of Canada through Genome Canada and the Canadian Institutes of Health Research, the Ministère de l'Économie, de la Science et de l'Innovation du Québec through Genome Québec, and the Quebec Breast Cancer Foundation; the National Cancer Institute Genetic Associations and Mechanisms in Oncology (GAME-ON) initiative and Discovery, Biology and Risk of Inherited Variants in Breast Cancer (DRIVE) projects (National Institutes of Health [NIH] grants U19 CA148065 and X01HG007492); and Cancer Research UK (C1287/A10118 and C1287/A16563). Genotyping of the iCOGS array was funded by the European Union (HEALTH-F2-2009-223175); Cancer Research UK (C1287/A10710), the Canadian Institutes of Health Research for the "CIHR Team in Familial Risks of Breast Cancer" program; and the Ministry of Economic Development, Innovation and Export Trade of Quebec (grant PSR-SIIRI-701). Combination of the GWAS data was supported in part by NIH Cancer Post-Cancer GWAS initiative grant U19 CA148065 (DRIVE, part of the GAME-ON initiative). All studies and funders are listed in ref. [60]. The WHI program is funded by the National Heart, Lung, and Blood Institute, National Institutes of Health, US Department of Health and Human Services through contracts N01WH22110, 24152, 32100–2, 32105–6, 32108–9, 32111–13, 32115, 32118–32119, 32122, 42107–26, 42129–32, and 44221. This manuscript was not prepared in collaboration with investigators of the WHI, has not been reviewed, and/or approved by the WHI, and does not

necessarily reflect the opinions of the WHI investigators or the NHLBI. The study sponsors had no role in the design of the study, the collection, analysis, or interpretation of the data, the writing of the manuscript, or the decision to submit the manuscript for publication. Open access funding provided by Karolinska Institute.

## Author contributions

F.G. and K.C. conceived and designed the project; K.C. acted as the principal investigator; P.H., M.E, M.G., Women's Health Initiative organized patient recruitment and sample collection; F.G. and W.H. analyzed the data; W.H., M.G., P.H., K.C. contributed to data interpretation; F.G. and K.C. wrote the manuscript with input from all authors. All authors approved of the final manuscript.

## Competing interests

The authors declare no competing interests.
