## [Transparent Peer Review File · Nature Communications]

Reviewers' comments:

Reviewer #1 (Remarks to the Author):

This is a case-case analysis of interval versus screen-detected breast cancers across three datasets: KARMA, LIBRO, and WHI. The main finding is the association between interval cancers and non-breast cancers diagnosed before and after the interval cancer. Women with interval cancers also tended to have a positive family history of non-breast tumors. The authors suggest that women with true interval cancers have increased "genetic load" predisposing them to other cancers, and this is probably explained by rare variants given that breast and non-breast genetic risk scores comprised of common variants were inversely associated with interval cancers.

Novelty/impact

1. The authors have previously shown that interval cancers are enriched for PTVs in moderate- and high-penetrance genes (Li Clin Cancer Res 2017; Li Cancer Res 2018). Though this manuscript's findings are interesting, they do not shed major light on the genetics of interval cancers and primarily show that the increased genetic load in these women may result in increased risk of non-breast tumors. The genetic component of the analysis recapitulates prior findings that a genetic risk score is inversely associated with interval cancers, while showing that non-breast genetics risk scores are not associated with interval cancers - an unsurprising finding given that overlapping breast variants were excluded from these risk scores. Given this analysis was on SNP genotyping data, the authors were unable to assess rare variants in this analysis (though this has been previously done in LIBRO, Li Cancer Res 2018). Thus, much remains to be addressed about this important question.

2. The potential clinical impact of the findings is also unclear. It is hard to see how these results could be leveraged for cancer risk assessment or prevention (either primary or secondary), assuming they are replicated elsewhere. For instance, the associations with colorectal and skin cancers are modest at best and would unlikely trigger changes in management - especially as we already screen for colorectal cancers while no skin cancer screening programs exist (likely due to it not meeting WHO criteria). Conversely, would the authors foresee intensive breast screening for lung cancer patients - a group enriched for other comorbidities?

3. Related to #2, several clarifications would improve the interpretability of the results around associations between interval cancers and non-BC tumors:

a) It is surprising that there was no association with ovarian cancer, given the link with BRCA1/2 mutations. Do the authors think this is a result of the analysis being underpowered, or are there alternative explanations?

b) Did skin cancers include melanomas, non-melanomas, or both? The implications would be different: again, the most obvious link would be between BRCA2 and melanoma (Goggins Int J Cancer 2004), though others have suggested that those with frequent basal cell cancers tend to be carriers of certain moderate/high-penetrance breast cancer mutations (Cho JCI Insight 2018).

c) Could the associations with bladder and lung cancer be due to the confounding effects of smoking? Is it possible to present smoking rates in women with interval vs screen-detected cancers?

4. The authors have previously shown that the breast cancer genetic risk score (GRS) is preferentially associated with screen-detected (rather than interval) cancers (Li Ann Onc 2016, Holm JCO 2015), and this analysis again found an inverse association between interval cancers and non-breast cancer GRS. This finding does not completely rule out the role of common variants in interval cancer risk. Two reasons for the inverse association between the GRS and interval cancers are: 1) the non-fitted GRS is preferentially associated with screen-detected cancers (which the authors acknowledge); 2) variants preferentially associated with interval cancers have not been identified, or could represent a subset of existing GWAS hits such as those associated with ER-negative cancers or other markers of unfavorable biology.

Analysis

The inclusion of WHI raises methodological concerns, as there are several features that make it quite different from LIBRO and KARMA. While the authors ameliorate this concern somewhat by presenting results by study in Table 1, there is still a concern about how accurately interval cancers were assessed, and if the relationship structure between risk factors and disease are truly similar in WHI. Overall, it would be reassuring to know if the main results held up, or showed similar directionality, with WHI excluded.

1. The authors ascertained interval cancers using a yearly questionnaire asking if a woman had undergone a mammogram since the last questionnaire, and by assuming the interceding mammogram happened in middle of the interval between surveys. To reduce the possibility of including screen-detected cancers, they only counted cancers diagnosed >3 months after the estimated prior mammogram as interval cancers. The authors acknowledge there are shortfalls to this method, and certainly Table 1 suggests it picked up a heterogeneous group of cancers. To help us better understand the limitations of this method:

- a) What is the average interval between responses on the screening history question, and were these questionnaires reliably completed annually? Was it possible for 2+ years to elapse between questionnaires, and if so were the authors able to determine if a woman had 1 or 2 mammograms in the intervening years (which would impact ability to accurately determine interval cancers)?
- b) What was the average screening interval and adherence rate to regular screening? In the U.S., screening is opportunistic and can happen at irregularly, thus the assumption that screening happened at the halfway point between two yearly questionnaires seems tenuous. It is totally plausible that many of these women were getting screened yearly given that guidelines from that era recommended annual screening - this would affect the ability to compare these interval cancers with those from LIBRO/KARMA, given that biology could be different. To that point, if WHI women were actually mostly getting yearly screening, it may actually be concerning that tumor characteristics were in fact similar to LIBRO/KARMA in Table 1.
- c) The authors mention the challenges of interval cancer ascertainment as a limitation, but more attention needs to be paid to this important consideration in the Discussion. Otherwise, supporting analyses or references should be provided to further assure us that their method was reliable.

2. Was breast density available for WHI? If the authors did not exclude cases with high breast density as they did with LIBRO/KARMA, this should be explicitly mentioned and discussed as limitation.

3. Why were only 971 genotypes available in WHI? Is this a random sample, or could they have differed somewhat from the overall cases? The authors should also point out which analyses included only 971 rather than 10,672 women from WHI in the captions or figure titles.

4. Another limitation is that WHI tended to be older women with greater all-cause mortality, thus limiting follow-up time to assess non-breast cancers following an interval cancer.

5. For the GRS analyses, it would also be interesting to see results with breast cancer variants included in the non-breast GRS's as it could reveal more about shared basis of risk between breast and non-breast cancers.

6. In Table 1 analysis, it appears that the logistic regression model did not adjust for density. Even though women with the very highest density were excluded, there is still a possibility of residual confounding. Also, how many patients with interval cancer (or what percentile) ended up being excluded based on breast density?

Specific items:

1. Please qualify wording in Introduction, paragraph 2 to clarify that not all interval cancer studies have showed increased mortality - see Houssami NPJ Breast 2017 for discussion of prognosis in

interval cancers

2. Results paragraph 3: wording suggests causality, please fix to "prior diagnosis of lung or bladder was associated with increased risk for interval cancer"

3. Please clarify which analyses excluded non-Caucasians - this is indicated in Table 1, but are non-Caucasians also excluded in Fig 2, Table S2? This should also be mentioned as a limitation.

4. Methods para 4: please provide some more background details on WHI: for example, it enrolled in the U.S. between years X and Y, with follow-up time of Z.

Reviewer #2 (Remarks to the Author):

This study investigated factors associated with interval breast cancers (BC) compared to those that were screen detected, with an emphasis on genetic factors. Using three independent studies totalling 14,846 women diagnosed with BC of which 1,772 were interval cancers, were more likely to have been diagnosed with a non-BC tumor before and after breast cancer diagnosis, are more likely to report a family history of non-BC tumors and have a lower genetic risk score for non-BC tumors. Interval breast cancers are more lethal and hence understanding factors for them is of great importance.

Questions I have are:

--Table 1 is a very nice summary of associations for various factors available in all three studies; however, i was surprised that they only emphasised statistical significance within individual studies and not across the studies. As this is probably one of the most comprehensive analyses of this nature, it would seem remiss not to meta-analyse or pool the results to determine the most consistent and significant factors.

--Related to above, while the focus was on genetic factors there are some non-genetic factors that are of interest and warrant some further thought in the discussion and limitations of the study; specifically education. Given that the top cancers that seem to be related to interval cancers are not hormonal in nature suggests different aetiologic factors (which was also evident in the tumor characteristics). Higher education is inversely associated with interval cancer diagnosis compared to screen detected cancers suggesting potentially lower ses, less likely to participate in screening or other comorbidities that might be associated with risk. I think this has to be discussed and do the authors have the opportunity to look at other factors such as smoking--a key risk factor for bladder and lung cancer to determine how other important factors might be influencing risk?

--Lastly, related to the risk factors as far as I can tell, no multivariable models were presented adjusting for all factors associated with risk--why is this?

--Given that other cancers beyond breast, and the most significant of which were bladder and lung cancer (which have few genetic markers associated and are more environmentally determined)--the authors should determine if smoking or other lifestyle genetic risk scores might be related--does not need to be all but at the very least obesity and smoking behaviours. Again, smoking seems to be a big confounder to the associations seen and should be investigated more closely to determine if its a driver of the associations seen.

Reviewer #1:

1. The authors have previously shown that interval cancers are enriched for PTVs in moderate- and high-penetrance genes (Li Clin Cancer Res 2017; Li Cancer Res 2018). Though this manuscript's findings are interesting, they do not shed major light on the genetics of interval cancers and primarily show that the increased genetic load in these women may result in increased risk of non-breast tumors. The genetic component of the analysis recapitulates prior findings that a genetic risk score is inversely associated with interval cancers, while showing that non-breast genetics risk scores are not associated with interval cancers - an unsurprising finding given that overlapping breast variants were excluded from these risk scores. Given this analysis was on SNP genotyping data, the authors were unable to assess rare variants in this analysis (though this has been previously done in LIBRO, Li Cancer Res 2018). Thus, much remains to be addressed about this important question.

We agree that further research is necessary to identify the underlying genetic basis since genetic mutations in known breast cancer genes would only account for a minor portion of the observed cases with two or more tumors. Our findings that other cancer risk scores in addition to the breast cancer score are mostly protectively associated with IC risk further underscores that we currently do not have sufficient understanding on how the risk for cancers overlaps and which mutations and genes impact both, risk and severity. Further large-scale sequencing efforts in patients with more than one tumor are thus necessary and unfortunately out of scope of the current publication. We have added this observation to the Discussion (first paragraph) and Conclusion section.

2. The potential clinical impact of the findings is also unclear. It is hard to see how these results could be leveraged for cancer risk assessment or prevention (either primary or secondary), assuming they are replicated elsewhere. For instance, the associations with colorectal and skin cancers are modest at best and would unlikely trigger changes in management - especially as we already screen for colorectal cancers while no skin cancer screening programs exist (likely due to it not meeting WHO criteria). Conversely, would the authors foresee intensive breast screening for lung cancer patients - a group enriched for other comorbidities?

The current study aimed rather at providing novel biological insights into interval cancer (genetics) and we cannot investigate and speculate too strongly on clinical impact. Most of the cases with prior tumors were diagnosed within one year after the last screening and thus would not necessarily benefit from yearly breast cancer screening intervals. Nevertheless, future studies are necessary to identify alternative

methods so that women with a prior tumor diagnosis and a family history of other tumors can be effectively screened for breast cancer. We have expanded on this topic in the Discussion section (sixth paragraph).

3. Related to #2, several clarifications would improve the interpretability of the results around associations between interval cancers and non-BC tumors:

a) It is surprising that there was no association with ovarian cancer, given the link with BRCA1/2 mutations. Do the authors think this is a result of the analysis being underpowered, or are there alternative explanations?

While we did find that interval cancer patients are more likely to be diagnosed with ovarian cancer after BC diagnosis, we indeed did not find an association with prior ovarian cancer and interval cancer status. While there is a well-documented genetic link between both cancers, those mutation would only account for fraction of the total diseases overlap and thus, we would need to investigate the association in familial and potentially younger cases. Due to the high mortality of ovarian cancer, only few cases with prior ovarian cancer are included in our study and thus we cannot perform such analyses. Alternatively, we hypothesize that removal of both the uterus and the ovaries/fallopian tubes as part of ovarian cancer treatment may change sex hormone levels favorably for interval cancer risk, which is likely linked to hormonal changes due to its risk association with hormone replacement therapy and/or BMI. We have further clarified this observation in the Discussion (second paragraph).

b) Did skin cancers include melanomas, non-melanomas, or both? The implications would be different: again, the most obvious link would be between BRCA2 and melanoma (Goggins Int J Cancer 2004), though others have suggested that those with frequent basal cell cancers tend to be carriers of certain moderate/high-penetrance breast cancer mutations (Cho JCI Insight 2018).

We thank the reviewer for their valuable comment and agree that we need to address this matter. Currently, both melanoma and non-melanoma patients were included in the skin cancer analysis, since the Women's Health Initiative only reported on melanoma outcome and not on other skin cancer types. We now also present the data separately for melanoma and non-melanoma skin cancers (and their respective genetic risk scores) and found that the increased number of skin cancer patients can be attributed to both non-melanoma skin tumors and to melanomas although the observed effect sizes for non-melanoma skin tumors are larger than the ones observed for melanoma. We have added this observation to the Results (third and fifth paragraph) and have updated Figure 1, Figure 2 and Supplementary Figure 2 to include both non-melanoma skin cancers and melanomas separately.

c) Could the associations with bladder and lung cancer be due to the confounding effects of smoking? Is it possible to present smoking rates in women with interval vs screen-detected cancers?

We agree that this is one potential explanation for some of the observed associations. We now also show the association of smoking status as well as other risk factors with interval breast cancer in a meta-analysis and have additionally included a multivariable analysis adjusted for known risk factors of interval cancer (Results, second paragraph).

4. The authors have previously shown that the breast cancer genetic risk score (GRS) is preferentially associated with screen-detected (rather than interval) cancers (Li Ann Onc 2016, Holm JCO 2015), and this analysis again found an inverse association between interval cancers and non-breast cancer GRS. This finding does not completely rule out the role of common variants in interval cancer risk. Two reasons for the inverse association between the GRS and interval cancers are: 1) the non-fitted GRS is preferentially associated with screen-detected cancers (which the authors acknowledge); 2) variants preferentially associated with interval cancers have not been identified, or could represent a subset of existing GWAS hits such as those associated with ER-negative cancers or other markers of unfavorable biology.

We agree with the reviewer that variants that are specifically associated with interval cancer have not been identified yet. Therefore, we now show the association summary statistics for the association of individual cancer variants with interval cancer risk in **Supplementary Table S1**. After adjustment for multiple testing, none of the variants remains statistically significantly associated with IC risk (Results, last paragraph). We also discuss those findings in the Discussion (fourth paragraph). Nevertheless, there may be additional variants associated specifically with IC risk, which can potentially be identified by GWAS. Such an effort is currently underway within the BCAC.

5. The inclusion of WHI raises methodological concerns, as there are several features that make it quite different from LIBRO and KARMA. While the authors ameliorate this concern somewhat by presenting results by study in Table 1, there is still a concern about how accurately interval cancers were assessed, and if the relationship structure between risk factors and disease are truly similar in WHI. Overall, it would be reassuring to know if the main results held up, or showed similar directionality, with WHI excluded.

In general, the association of prior and subsequent tumors with IC status as well as the association of the genetic scores is consistent across three cohorts. We have now also included histological grade as an additional tumor characteristic and again found consistent orientation of the effect sizes (**Table 1**). Nevertheless, we now also show the association of other tumors as well as the cancer genetic scores with IC risk without WHI and found similar results (new **Supplementary Figure 3**). We also mention those points now in the Discussion (fifth paragraph).

6. The authors ascertained interval cancers using a yearly questionnaire asking if a woman had undergone a mammogram since the last questionnaire, and by assuming the interceding mammogram happened in middle of the interval between surveys. To reduce the possibility of including screen-detected cancers, they only counted cancers diagnosed >3 months after the estimated prior mammogram as interval cancers. The authors acknowledge there are shortfalls to this method, and certainly Table 1 suggests it picked up a heterogeneous group of cancers. To help us better understand the limitations of this method:

a) What is the average interval between responses on the screening history question, and were these questionnaires reliably completed annually? Was it possible for 2+ years to elapse between questionnaires, and if so were the authors able to determine if a woman had 1 or 2 mammograms in the intervening years (which would impact ability to accurately determine interval cancers)?

We agree with the reviewer that our approach has some shortcomings regarding accuracy of the interval cancer status. Therefore, we now also report additional metrics related to the estimated screening regime in the WHI in the Methods section (eighth paragraph). In particular, we computed the average time between screening and diagnosis (which was around 9-10 months, i.e. most IC cases are first year cases), average interval between screening times in WHI (around 19 months, as expected from a population screened both annually and bi-annually) as well as number of BC diagnosis per 10,000 mammograms in order to allow comparison with KARMA, LIBRO1 as well as other screening-based studies. In general, the metrics of the self-reported mammograms is comparable to the metrics observed in population with actual screening data. The yearly questionnaire was answered fairly regularly for WHI participants, with an average time between questionnaires (for all participants) of around 380 days. We have included those measures in the Methods section (eighth paragraph) to assure the reader that our computed IC status should be quite accurate.

b) What was the average screening interval and adherence rate to regular screening? In the U.S., screening is opportunistic and can happen at irregularly, thus the assumption that screening happened at the halfway point between two yearly questionnaires seems tenuous. It is totally plausible that many of these women were getting screened yearly given that guidelines from that era recommended annual screening - this would affect the ability to compare these interval cancers with those from LIBRO/KARMA, given that biology could be different. To that point, if WHI women were actually mostly getting yearly screening, it may actually be concerning that tumor characteristics were in fact similar to LIBRO/KARMA in Table 1.

Previous research has shown that there is little difference between interval tumors detected within the first or second year of the screening regimes. Thus, we proposed that even interval cancer patients which were screened yearly will have similar tumor characteristics as those that are screened every two years. We now mention this observation as well as the average time between screening and interval cancer diagnosis in the Methods section (eighth paragraph).

c) The authors mention the challenges of interval cancer ascertainment as a limitation, but more attention needs to be paid to this important consideration in the Discussion. Otherwise, supporting analyses or references should be provided to further assure us that their method was reliable.

We agree that our method to compute interval cancer status from questionnaire is not established and has several limitations to the classical approach using screening dates (with screen results). Since such an approach, to our knowledge, has not been performed before, we wanted to be sure that our results hold up even without the WHI. Therefore, we now present the main results also without individuals from the WHI (see above and the new Supplementary Figure 3) and found similar results.

7. Was breast density available for WHI? If the authors did not exclude cases with high breast density as they did with LIBRO/KARMA, this should be explicitly mentioned and discussed as limitation.

Agreed and done (see Discussion, fifth paragraph).

8. Why were only 971 genotypes available in WHI? Is this a random sample, or could they have differed somewhat from the overall cases? The authors should also point out which analyses included only 971 rather than 10,672 women from WHI in the captions or figure titles.

In general, less than 10% of the BC patients were genotyped in WHI and had their genotypes deposited in dbGAP. The patients were part of different genotyping runs as part of different sub studies and thus appear to not be specifically selected. We agree that it is important to mention the actual number of patients with genotype data in the Figures and now mention in the precise number of samples included in the respective analyses in the figure legends. We also now point out that the breast cancer samples were genotyped as part of different sub-studies in Methods section (11th paragraph).

9. Another limitation is that WHI tended to be older women with greater all-cause mortality, thus limiting follow-up time to assess non-breast cancers following an interval cancer.

We agree with this statement. The increased all-cause mortality and thus shorter follow-up time after BC diagnosis in the older WHI population may in fact explain the reduced effect size for tumor after BC diagnosis seen in **Table 1**. We have added this observation to the Discussion section (fifth paragraph).

10. For the GRS analyses, it would also be interesting to see results with breast cancer variants included in the non-breast GRS's as it could reveal more about shared basis of risk between breast and non-breast cancers.

The current correlation analyses were provided to assure the reader that our other cancer scores are indeed not correlated to the breast cancer score and thus provide novel associations with interval cancer risk. While we agree that there is merit in studying the genetic correlation between various cancers, other groups have already published such analyses (see *Quantifying the Genetic Correlation between Multiple Cancer Types* by Lindström *et al.* 2017 and various other groups). Finally, the correlations are computed in breast cancer patients and thus may exhibit specific genetic correlations which will not hold true in the general population. As suggested, we have computed the pairwise correlation between all cancer GRS including variants in breast cancer loci (see attached Figure below) and found that the breast cancer GRS is significantly positively correlated to the skin cancer GRS and negatively correlated to the bladder, CNS, lung and CRC GRS, which recapitulates prior findings.

Still, we would prefer not to include this analysis in our current manuscript since it may distract from our main findings and does not necessarily provide novel insights in addition to those that were previously published by other groups. Furthermore, the correlation of other genetic scores including variants in known breast cancer loci with the breast cancer GRS makes it far more difficult to interpret our results.

11. In Table 1 analysis, it appears that the logistic regression model did not adjust for density. Even though women with the very highest density were excluded, there is still a possibility of residual confounding. Also, how many patients with interval cancer (or what percentile) ended up being excluded based on breast density?

We now performed an additional analysis using multivariable logistic regression including percent mammographic density as a covariable and found virtually no changes in the observed main associations, except the influence of exclusion of the patients from the WHI. We now mention this analysis in the Results section (second paragraph). We also now mention the percentile of IC patients excluded based on mammographic density in the Methods section (sixth paragraph).

Reviewer #2:

1. Table 1 is a very nice summary of associations for various factors available in all three studies; however, I was surprised that they only emphasised statistical significance within individual studies and not across the studies. As this is probably one of the most comprehensive analyses of this nature, it would seem remiss not to meta-analyse or pool the results to determine the most consistent and significant factors.

We agree and have expanded **Table 1** to include smoking status as well as a meta-analysis of the risk factors. We also present those results in more detail in the Results (first paragraph).

2. Related to above, while the focus was on genetic factors there are some non-genetic factors that are of interest and warrant some further thought in the discussion and limitations of the study; specifically, education. Given that the top cancers that seem to be related to interval cancers are not hormonal in nature suggests different aetiologic factors (which was also evident in the tumor characteristics). Higher education is inversely associated with interval cancer diagnosis compared to screen detected cancers suggesting potentially lower ses, less likely to participate in screening or other comorbidities that might be associated with risk. I think this has to be discussed and do the authors have the opportunity to look at other factors such as smoking--a key risk factor for bladder and lung cancer to determine how other important factors might be influencing risk?

We thank the reviewer for this comment and have include smoking as well as a meta-analysis of risk factors and tumor characteristics in **Table 1** (see above).

We agree that education attainment is an interesting risk factor, however we do not think that it is associated with IC risk due to screening adherencesince analysis of interval cancers inherently excludes women without prior screening or those who missed their prior scheduled mammogram. Rather, education attainment may be a marker for other socio-economic and/or health related outcomes, which require further investigation and are slightly out of scope of the current study.

3. Lastly, related to the risk factors as far as I can tell, no multivariable models were presented adjusting for all factors associated with risk---why is this? Given that other cancers beyond breast, and the most significant of which were bladder and lung cancer (which have few genetic markers associated and are more environmentally determined) -- the authors should determine if smoking or other lifestyle genetic risk scores might be related -- does not need to be all but at the very least obesity and smoking behaviours. Again, smoking seems to be a big confounder to the associations seen and should be investigated more closely to determine if it is a driver of the associations seen.

We agree and now present the results additionally adjusted for all factors significantly associated with interval cancer risk (and also percent mammographic density, as suggested by Reviewer #1, see Results second paragraph). In principle, adjustment for those factors did not change the observed associations and should also capture unknown genetic factors that may confound or influence the analysis. Nevertheless, as suggested, we computed genetic scores for five traits (BMI, ever smoking, education attainment, age at first birth and age at menarche) and found that none of those were statistically significantly associated with IC risk (see attached Figure below). Since the current focus of this study was to elucidate the role of other cancers in interval cancer risk we think investigating other risk scores and their association with IC is out of scope and may distract from the main finding that the presence of other tumors is more common in interval than in screen detected patients.

Reviewers' comments:

Reviewer #1 (Remarks to the Author):

The authors have been quite responsive to concerns raised on the first round of review. In particular, they have provided more extensive supporting information on their method of interval cancer ascertainment in WHI and analyses showing that excluding WHI did not substantively change findings. Have also revised analyses to distinguish between melanoma and non-melanoma skin cancers.

A few more comments about Table 1:

a) Authors need to clarify how some of these analyses were performed. They present logistic regression results for several variables that are continuous, such as tumor size and number of positive lymph nodes. How are these results to be interpreted?

b) One nagging thought about generalizability of findings: the interval cancers found here, especially from KARMA and WHI, have strikingly high percentage of ER positivity and low percentage of distant metastases, HER2 positivity, and triple negativity compared with other studies (even the Holm JCO 2015 study which ostensibly covered a similar population). What is the explanation for this? Did the excluded interval cancers (those occurring in dense breasts) differ in biology? Can the authors present descriptive characteristics of these cancers?

The Discussion is considerably improved after revision. However, though the authors spend considerable amount of time addressing what's not going on (known SNPs are not responsible for increased risk of non-BC cancers and/or family history of non-BC cancers), they dedicate relatively little time to discussing what they think is going on. This could go after 4th paragraph.

To make room for above, could cut some discussion about clinical implications as several of them seem to be a stretch based on findings here.

a) "A prior non-BC tumor diagnosis in women in addition to a positive family history of breast cancer or other tumors could encourage preventive measures such as...prophylactic medication such as low dose or topical tamoxifen." As authors mention in the introduction, ER-negative cancers are more common within interval cancers vs screen-detected, so hormonal chemoprevention would be of doubtful benefit.

b) "Novel screening regimes with alternative methods and schedules are warranted for cancer survivors" is a sweeping statement and should be qualified. First, none of the associations detected here are arguably strong enough to change practice, even if replicated. Second, this statement would be true only for certain cancers since associations were only detected for several. Optimally, intensive screening would also be based on other risk factors and consideration of prognosis/comorbidities.

I think it's fine to caution that this paper is intended to describe genetic risk around interval cancers, and results are not intended to change clinical practice beyond raising awareness that some of these patients and family members could be at risk of other cancers. While mentioning caveats that other considerations like accompany risk factors, comorbidities, prognosis, etc would also need to be considered in management.

Other issues

1. Please qualify wording in Introduction, paragraph 2 to clarify that not all interval cancer studies have showed increased mortality.

2. Results paragraph 3: wording suggests causality, please fix to "prior diagnosis of X cancers... was associated with increased risk for interval cancer." Also, "increased risk due to skin cancer can be attributed to both melanoma and non-melanoma skin cancers" is debatable as melanoma

association did not reach statistical significance.

3. Please clarify which analyses excluded non-Caucasians - this is indicated in Table 1, but are non-Caucasians also excluded in Fig 2, Table S2? This should also be mentioned as a limitation.

4. Discussion paragraph 1: second sentence is confusing. I think you meant to say: "we demonstrated that in patients with a second primary tumor diagnosis besides interval breast cancer were more likely to have a family history of non-breast cancer."

5. Methods paragraph 4: please provide some more background details on WHI: for example, it enrolled in the U.S. between years X and Y, with follow-up time of Z. Please also include relevant citations.

6. Methods paragraph 6: you excluded KARMA patients with "medium or high" mammographic density while you excluded LIBRO-1 patients with "dense breasts." Please clarify the exact criteria used to exclude women based on density, were they the same?

7. Methods paragraph 8: please fix to "biennially" to mean every other year. "Bi-annually" means twice yearly.

8. Methods paragraph 12: which reference was WHI imputed to? Was it also 1000 Genomes?

Reviewer #2 (Remarks to the Author):

The authors have satisfactorily addressed my comments.

Reviewer #1:

1. A few more comments about Table 1:

a) Authors need to clarify how some of these analyses were performed. They present logistic regression results for several variables that are continuous, such as tumor size and number of positive lymph nodes. How are these results to be interpreted?

The odds ratio represents the increased odds per unit, i.e. per cm of tumor size or per additional lymph node. We have further clarified this in the foot notes of **Table 1**.

b) One nagging thought about generalizability of findings: the interval cancers found here, especially from KARMA and WHI, have strikingly high percentage of ER positivity and low percentage of distant metastases, HER2 positivity, and triple negativity compared with other studies (even the Holm JCO 2015 study which ostensibly covered a similar population). What is the explanation for this? Did the excluded interval cancers (those occurring in dense breasts) differ in biology? Can the authors present descriptive characteristics of these cancers?

Previous research showed that there are differences in tumor characteristics between high and low dense IC cancers. This is particularly the case because breast cancer in women with high breast density is frequently missed at the prior screening due to masking and thus those tumors more closely resemble screen detected cancers (see Holm et al. 2015 JCO).

The frequency of ER positive tumors in Swedish interval cancer populations is around 80%, in line with the frequencies observed by other studies (see citations ³⁹⁻⁴¹ and Holm et al. 2015 JCO, Supplementary Table 1). Overall, the slightly higher frequency of ER positive tumors in KARMA and WHI can be attributed to different periods of inclusion, increased age and recruitment strategy. Adjusting the association of prior and subsequent tumors with interval cancer status for all tumor characteristics provided in Table 1 or restricting the analyses to patients with ER positive tumors only marginally altered the observed effect sizes and statistical significance.

Investigating the tumor characteristics of IC is not the main focus of this manuscript and were only presented to show that the IC tumors behave similarly across all studies. Indeed, the 95% confidence intervals of the association of the tumor characteristics with IC usually overlap between the three studies. Thus, the main conclusions of our manuscript should not be affected by those differences. We have added those differences in tumor characteristics compared to other studies as a potential limitation to our discussion to inform the reader (Discussion, last paragraph):

In addition, the frequency of estrogen receptor positive tumors is somewhat higher in our IC cases compared to the frequencies observed in IC patients from other studies ³⁹⁻⁴¹, which can be attributed to different periods of inclusion and differences in recruitment strategy. However, those differences should not affect the main conclusions of this study. Finally, in this study we have not identified the actual cause of the increased frequency of other tumors in interval compared to screen detected patients and thus were only able to speculate on potential mechanisms.

2. The Discussion is considerably improved after revision. However, though the authors spend considerable amount of time addressing what's not going on (known SNPs are not responsible for increased risk of non-BC cancers and/or family history of non-BC cancers), they dedicate relatively little time to discussing what they think is going on. This could go after 4th paragraph.

We agree that this is a limitation of our study since we have not identified the actual cause of the increased frequency of other tumors in interval compared to screen detected breast cancers and have added this to the limitations section in the Discussion (last paragraph). In addition, we have expanded the discussion about the contribution of rare mutations to the observed associations in the second paragraph:

Our results are in line with recent findings that interval breast cancer patients compared to screen-detected patients have fewer common breast cancer risk increasing alleles and, on the other hand, are more likely to carry rare deleterious (loss of function) mutations in known or suspected cancer genes^{17, 18}. While previous efforts to identify cancer genes have largely focused on single tumor types, recent evidence suggests a pleiotropic role those genes²¹ due to their involvement in fundamental processes required for tumor initiation and propagation such as genomic integrity^{22, 23}, hormonal processes, angiogenesis and inflammation²⁴. As such, mutations in those genes frequently predispose for different cancers and cancer syndromes^{21, 25} and might result in multiple tumor diagnoses over an individual's lifetime as well as increase familial risk for tumors. Importantly, patients with germline mutations in cancer genes are often diagnosed with more aggressive and fast growing tumors²⁶⁻²⁸ and thus may more likely be diagnosed with interval than screen detected breast cancer. Nevertheless, mutations in currently known cancer genes can only explain a small portion of the observed association. Thus, further large-scale sequencing efforts in patients with multiple tumors are necessary to uncover the genetic basis for both, multiple cancer diagnoses and increased interval breast cancer risk.

Apart from the contribution of rare mutations to the observed association, we prefer not to speculate on other factors that may be involved in the current study. It is plausible that cancers survivors for instance may have an immune system that is more attuned to the detection and elimination of tumor cells and thus the newly developing tumor needs to be fast growing, potentially with low immunogenicity to avoid the increased immune surveillance and would therefore be more likely be detected between two screens. However, we currently do not have any data to support this claim and we therefore want to investigate the precise role of other tumors as well as the timing of prior and subsequent tumors in breast cancer in a separate, larger study.

a) **“A prior non-BC tumor diagnosis in women in addition to a positive family history of breast cancer or other tumors could encourage preventive measures such as...prophylactic medication such as low dose or topical tamoxifen.”** As authors mention in the introduction, ER-negative cancers are more common within interval cancers vs screen-detected, so hormonal chemoprevention would be of doubtful benefit.

We agree that the increased frequency of ER/PR negative cancers in IC would reduce the effectiveness of hormonal prevention. However, the majority of IC cases would still be either PR or ER positive and thus would potentially profit from such an intervention. We have shortened this paragraph as suggested by the reviewer (see answer below)

b) **“Novel screening regimes with alternative methods and schedules are warranted for cancer survivors”** is a sweeping statement and should be qualified. First, none of the associations detected here are arguably strong enough to change practice, even if replicated. Second, this statement would be true only for certain cancers since associations were only detected for several. Optimally, intensive screening would also be based on other risk factors and consideration of prognosis/comorbidities.

I think it's fine to caution that this paper is intended to describe genetic risk around interval cancers, and results are not intended to change clinical practice beyond raising awareness that some of these patients and family members could be at risk of other cancers. While mentioning caveats that other considerations like accompany risk factors, comorbidities, prognosis, etc would also need to be considered in management.

We completely agree with the reviewer and have changed this section appropriately as suggested. The current section has been shorted as follows:

Thus, the current results are not intended to be directly relevant for clinical practice beyond raising awareness that some of these patients and family members are at increased risk for interval breast cancer and other tumors. While the increased risk for interval cancer in cancer survivors may have potential implications for screening or prevention programs, other risk factors, co-morbidities as well as their potential impact on prognosis need to be considered jointly in disease management.

3. Please qualify wording in Introduction, paragraph 2 to clarify that not all interval cancer studies have showed increased mortality.

We agree that not all studies have found an increased mortality, particularly earlier studies. We have reworded this section and the section now states:

Compared to breast cancers detected at a screening, interval cancers are characterized by more aggressive tumor characteristics and poorer prognosis (i.e. increased mortality) in most recent studies (summarized in ⁶).

4. Results paragraph 3: wording suggests causality, please fix to “prior diagnosis of X cancers... was associated with increased risk for interval cancer.”

Agreed. This section has been reworded to the following:

We found that a prior diagnosis of lung (OR: 2.57, 95% CI: 1.09-6.06, $P < 0.05$), colorectal (OR: 1.74, 95% CI: 1.02-2.97, $P < 0.05$) or skin (OR: 1.70, 95% CI: 1.04-2.78, $P < 0.05$) cancer was significantly associated with increased risk for interval breast cancer (Figure 1).

5. Also, “increased risk due to skin cancer can be attributed to both melanoma and non-melanoma skin cancers” is debatable as melanoma association did not reach statistical significance.

We agree and now mention that melanoma individually was not statistically significantly associated with IC risk (Results, third paragraph):

The increased risk due to skin cancer can be attributed to both, melanoma and non-melanoma skin cancers, although the association was not statistically significant for the latter.

6. Please clarify which analyses excluded non-Caucasians - this is indicated in Table 1, but are non-Caucasians also excluded in Fig 2, Table S2? This should also be mentioned as a limitation.

Only the genetic analyses and analyses adjusted for any GRS were restricted to Caucasians. We have added this to the Figure and Table legends/foot notes where appropriate. Restricting the genetic analyses to Caucasians in WHI is warranted in order to avoid confounding our analyses due to population stratification.

7. Discussion paragraph 1: second sentence is confusing. I think you meant to say: “we demonstrated that in patients with a second primary tumor diagnosis besides interval breast cancer were more likely to have a family history of non-breast cancer.”

Agreed. We have changed this sentence accordingly:

In this study, we showed that non-BC tumors, either before or after BC diagnosis, are associated with interval breast cancers and that in breast cancer patients with a second primary tumor diagnosis, interval compared to screen-detected patients are more likely to report a family history of other cancers. In addition, we demonstrated that interval cancer patients compared to screen-detected patients have fewer cancer risk increasing alleles of both, breast cancer and other cancers.

8. Methods paragraph 4: please provide some more background details on WHI: for example, it enrolled in the U.S. between years X and Y, with follow-up time of Z. Please also include relevant citations.

Agreed and done. We now state:

The WHI is a prospective national health study of more than 160,000 postmenopausal women aged 50–79 recruited between 1993 and 1998 at 40 clinical sites in the U.S.^{44, 45}, with an average follow-up time of 17.20 (S.D. 4.17) years.

9. Methods paragraph 6: you excluded KARMA patients with “medium or high” mammographic density while you excluded LIBRO-1 patients with “dense breasts.” Please clarify the exact criteria used to exclude women based on density, were they the same?

We used the same criteria for both studies and have clarified this in the Methods section

10. Methods paragraph 8: please fix to “biennially” to mean every other year. “Bi-annually” means twice yearly.

Agreed and done

11. Methods paragraph 12: which reference was WHI imputed to? Was it also 1000 Genomes?

We agree that this information was missing and have added it to the Methods section:

Patients from the WHI were genotyped on different genotyping platforms as part of various sub-studies and were jointly imputed to the 1000 Genomes Phase 3 reference by the WHI investigators.

REVIEWERS' COMMENTS:

Reviewer #1 (Remarks to the Author):

The authors have satisfactorily addressed my prior critiques.